# Speckle-Based Transmission and Dark-Field Imaging for Material Analysis with a Laboratory X-Ray Source

**DOI:** 10.3390/s25082581

**Published:** 2025-04-19

**Authors:** Diego Rosich, Margarita Chevalier, Tatiana Alieva

**Affiliations:** 1Physics Institute of Cantabria (IFCA-CSIC-UC), Av. de los Castros s/n, 39005 Santander, Spain; 2Department of Radiology, Physiotherapy and Rehabilitation, Faculty of Medicine, Complutense University of Madrid, Pl. de Ramón y Cajal s/n, 28040 Madrid, Spain; chevalier@med.ucm.es; 3Department of Optics, Faculty of Physics, Complutense University of Madrid, Pl. de las Ciencias 1, 28040 Madrid, Spain; talieva@ucm.es

**Keywords:** X-ray speckle-based imaging, laboratory source, dark-field imaging, multimodal X-ray imaging, material analysis

## Abstract

Multimodal imaging is valuable because it can provide additional information beyond that obtained from a conventional bright-field (BF) image and can be implemented with a widely available device. In this paper, we investigate the implementation of speckle-based transmission (T) and dark-field (DF) imaging in a laboratory X-ray setup to confirm its usefulness for material analysis. Three methods for recovering T and DF images were applied to a sample composed of six materials: plastic, nylon, cardboard, cork, expanded polystyrene and foam with different absorption and scattering properties. Contrast-to-noise ratio (CNR) and linear attenuation, absorption and diffusion coefficients obtained from BF, T and DF images are studied for two object-to-detector distances (ODDs). Two analysis windows are evaluated to determine the impact of noise on the image contrast of T and DF images and the ability to retrieve material characteristics. The unified modulated pattern analysis method proves to be the most reliable among the three studied speckle-based methods. The results showed that the CNR of T and DF images increases with larger analysis windows, while linear absorption and diffusion coefficients remain constant. The CNR of T images decreases with increasing ODD due to noise, whereas the CNR of DF images exhibits more complex behaviour, due to the material-dependent reduction in DF signal with increasing ODD. The experimental results on the ODD dependence of T and DF signals are consistent with recently reported numerical simulation results of these signals. The absorption coefficients derived from T images are largely independent of the ODD and the speckle-based method used, making them a universal parameter for material discrimination. In contrast, the linear diffusion coefficients vary with the ODD, limiting their applicability to specific experimental configurations despite their notable advantages in distinguishing materials. These findings highlight that T and DF images obtained from a laboratory X-ray setup offer complementary insights, enhancing their value for material analysis.

## 1. Introduction

In the last decade, the growing interest in the implementation of multimodal X-ray imaging related to absorption, small-angle scattering and refraction phenomena has focused on speckle-based imaging (SBI) methods [1,2,3]. This is due to their relative simplicity, their cost effectiveness and the availability of the required experimental setup, in contrast to interferometric methods, as well as the potential transferability of the results to various applications [4,5,6]. SBI methods consist in analyzing the modulation of the near-field speckle pattern produced by a diffuser when a sample is incorporated into the X-ray beam. There are several methods for retrieving transmission (T), dark-field (DF) and phase contrast (PC) images associated with absorption, small-angle scattering and refraction events, respectively, from diffuser alone (reference image) and diffuser with sample (speckled image) image pairs. The X-ray speckle tracking method (XST), X-ray speckle scanning method (XSS) [1,7,8,9] and the unified modulated pattern analysis method (UMPA) [10] are the most used ones for multimodal imaging. XST and XSS perform cross-correlation analysis between the reference and speckled sample images, while UMPA is based on the minimization of a cost function. XSS and UMPA are used in a diffuser scanning modality, while XST is applied for the single-shot case (only one pair of images is required). All these methods have their own set of pros and cons. On the one hand, while XST requires only two acquisitions to provide multimodal images, it suffers from low spatial resolution and excessive noise. On the other hand, XSS and UMPA have an increased spatial resolution, but they require the acquisition of several reference and speckled sample images (usually up to 10–20), which increases both acquisition and processing times, as well as the incorporation of a motorized translation stage. In addition, these methods can result in higher radiation doses than in the case of XST. The advantages of one-dimensional (1D) diffuser scanning methods in comparison with 2D diffuser scanning in a plane transverse to the beam direction are their easy implementation in a cost-efficient laboratory setup as well as less acquisition time. These advantages may compensate for the demonstrated superior performance of 2D methods [11].

Multimodal imaging becomes useful when it offers additional insights about a sample beyond its conventional X-ray image, which is referred to hereafter as the bright-field (BF) image. BF images contain mixed information about different physical processes involved in radiation–matter interactions (absorption, scattering and refraction) that cannot be directly separated. The aforementioned methods (XST, XSS and UMPA) enable the recovery of PC, T and DF images. However, the phase information can be recovered from single-shot defocused BF images [12,13]. A comparison of this approach with SBI methods for PC imaging using a laboratory X-ray source was conducted in our recent work [14]. This study is specifically focused on the analysis of T and DF images. The former provides information about sample absorption, whereas the latter gives insights into sample features that fall below the system’s resolution. DF, T and BF images will be compared based on quantitative analysis instead of visual inspection, as in earlier publications. In recent speckle-based imaging studies, like the one in reference [15], linear attenuation and diffusion coefficients are used for material characterization. The linear diffusion coefficient has been previously introduced for material scattering characterization for quantitative X-ray DF computed tomography using a grating interferometer [16] and edge illumination [17]. In particular, in [16], it has been shown experimentally that the DF signal decreases with sample thickness, as does the T signal. However, the variation in such signals with the ODD has not been addressed. Here, we additionally employ the contrast-to-noise ratio for the quantitative analysis of DF, T and BF images.

The goal of this paper is to demonstrate the benefits of speckle-based multimodal (T and DF) imaging for material analysis compared to BF imaging. This research was conducted using a standard laboratory X-ray setup with a polychromatic incoherent source and an imaging system with a detector with approximately 10 µm resolution. Furthermore, the study aims to analyze the differences in T and DF image recovery using different methods (XST, XSS and UMPA) and analysis windows, as well as to verify the results of numerical simulations reported in reference [18]. Measurements at two different object-detector-distances were performed with this objective.

This paper is organized as follows. In Section 2, we describe the characteristics of the used X-ray setup, the image acquisition conditions and the sample and briefly discuss the fundamentals behind and implementation of XST, XSS and UMPA methods. In Section 3, we present and discuss the experimental results. The article concludes with Section 4, which summarizes the main findings derived from the analysis.

## 2. Materials and Methods

### 2.1. Experimental Setup and Image Acquisition Conditions

The experimental laboratory setup used in this study is presented in Figure 1a. It consists of a microfocus X-ray source mod. L10951-04 from Hamamatsu (Hamamatsu Photonics K.K., Shizuoka, Japan) with a tungsten target and an imaging system (AA60 M11427-62, Hamamatsu Photonics K.K, Shizuoka, Japan) comprising a 10 µm thick scintillator layer of P43 phosphor (Gd_2_O_2_S:Tb) optically coupled to an ORCA Flash 4.0 V2 CMOS camera (Hamamatsu Photonics K.K, Shizuoka, Japan). The complete detector arrangement has an array of 2048 × 2048 pixels with a pixel size of 13.5 µm. The experiments were conducted at a voltage of 50 kV and a tube current of 120 µA for achieving the smallest possible source focal spot size of 20 µm, resulting in an X-ray beam with a mean energy of 12 keV. Previous studies have shown that the spatial coherence of the X-ray beam—defined by the source size and sample position—is not a restrictive factor for speckle-based methods [1,2,7]. However, in our case, the smallest source size, measuring 20 μm compared to the detector pixel size of 13.5 μm, reduces the diffuser blurring and increases the visibility. Achieving a smaller source size requires reducing the beam power, and consequently the tube current (μA). To offset the resulting low X-ray fluence, increasing the exposure time becomes necessary, which in our case is the maximum possible 10 s, for all acquired images, including the BF images.

The source–diffuser and the diffuser–object distances were 400 mm and 200 mm, respectively. Two object-to-detector distances (ODDs) of 140 mm and 340 mm were considered. The corresponding object magnifications are 1.23 and 1.57.

The diffuser was prepared according to the most common procedure consisting of stacking several diffuser layers [1,15]. It was made with three layers of silicon carbide sandpaper (P600) with an average grit size of 25.8 µm. The speckle sizes in the detector plane were approximately 3.7 and 4.5 pixels in size for ODDs equal to 140 mm and 340 mm, respectively. The visibility of the generated speckle pattern (see Figure 1b) was calculated as the average of the ratios of the standard deviation and mean intensity within a subset of several windows of 100 × 100 pixels in size. The visibility value was approximately 12% for both ODDs. The diffuser was moved in the x-direction, transverse to the beam propagation (Figure 1a), using a motorized translation stage mod. LSQ075A-T3A from Zaber (Zaber Technologies, Vancouver, BC, Canada) with a repeatability of less than 2 µm. It was displaced in steps of 50 µm, corresponding to 92.5 µm and 117.5 µm in the detector plane for ODDs of 140 mm and 340 mm, respectively. This step size represents the smallest increment permitted by our experimental setup. A total of 15 pairs of reference (diffuser only) and speckled sample (sample with diffuser) images were acquired as well as a BF image of the sample for each ODD. Three exposures were taken at each step for both the diffuser and speckled image that were averaged before being processed. Dark-current and flat-field (20 images each) corrections were applied to all the images before processing.

### 2.2. Sample Arrangement

We analyzed a sample comprising six materials: plastic, probably polyethylene (1), nylon (2), cardboard (3), cork (4), expanded polystyrene (5) and foam (6), each with different absorption and scattering properties. The sample shown in the photograph in Figure 2a closely resembles the one used in the experiments; however, the objects are slightly displaced from their original positions. The thicknesses of the materials measured with a calliper are displayed in Table 1.

### 2.3. Imaging Modalities Description

#### 2.3.1. X-Ray Speckle Tracking and 1D-Scanning Methods

In the case of XST, the DF image is defined as the ratio of the visibilities of the speckled sample ($v_{S}$) and the reference sample ($v_{R}$) images inside an analysis window,(1)$$DF\left(r\right)=\frac{v_{S}\left(r\right)}{v_{R}\left(r\right)},$$
where $r=\left(x,y\right)$ is the position vector. The visibilities of both images are computed according to(2)$$v\left(r\right)=\frac{\sigma \left(r\right)}{\bar{I}\left(r\right)},$$
where $\bar{I}\left(r\right)$ and $\sigma \left(r\right)$ are the mean pixel value and the standard deviation in an analysis window centred in the pixel located at r. The transmission image associated with the absorption phenomena is obtained as the ratio of the mean intensities of the speckled sample ($\bar{I_{S}}$) and reference $\left(\bar{I_{R}}\right)$ images inside the analysis window:(3)$$T\left(r\right)=\frac{\bar{I_{S}}\left(r\right)}{\bar{I_{R}}\left(r\right)}.$$

All the considered methods use a sliding window that moves across each pixel in the image. As a result, they produce images which have the same number of pixels as in the input one. In XST, this analysis window is squared, meaning that every pixel in the reconstructed image is calculated using the surrounding *M* × *M* pixel region. The window size is usually chosen as a trade-off between spatial resolution, overall noise level in the reconstructed images and the speckle size at the detector plane. Here, we consider two analysis windows with *M* = 5 and *M* = 11 pixels.

The XSS method was proposed for improving the spatial resolution of the retrieved DF and T images [1]. In this case, the diffuser is shifted at regularly spaced steps in a perpendicular direction to that of the X-ray beam (horizontal in our experiments) with and without the sample inserted in the beam. Then, two sets of *N* = 15 images are obtained. The analysis window for XSS is constructed in the following way. For each pixel, *M* pixels are selected in the perpendicular direction of the scan, forming in an *M* × 1 array. This is carried out for the same pixel across all images in the scan. These arrays are then concatenated, resulting in an *M* × *N* analysis window.

#### 2.3.2. Unified Modulated Pattern Analysis Method

The UMPA method is a scanning method in which the diffuser is shifted at *N*, in general, random positions rather than in regularly spaced steps along a single direction. However, in our case, regularly spaced steps were considered. The reconstruction is carried out by a least square minimization process using the cost function in Equation (4) not only at each pixel, but also at each diffuser step:(4)$$L=\overset{N}{\underset{n=1}{\sum }}\overset{\left(M-1\right)/2}{\underset{i=-\left(M-1\right)/2}{\sum }}\overset{\left(M-1\right)/2}{\underset{j=-\left(M-1\right)/2}{\sum }}w\left(x_{i},y_{j}\right){\left\{I_{Sn}\left(x_{i},y_{j}\right)-T\left(x_{i},y_{j}\right)\left[\bar{I_{R}}+DF\left(x_{i},y_{j}\right)\left(I_{Rn}\left(x_{i}+u_{x},y_{j}+u_{y}\right)-\bar{I_{R}}\right)\right]\right\}}^{2}.$$

Here, $I_{Rn}$ and $I_{Sn}$ are the intensities at pixel $\left(x_{i},y_{j}\right)$ of the *n*-th reference and speckle sampled image, respectively, $\bar{I_{R}}$ is the average intensity of the *N* reference images, and *M* is the size in pixels of the analysis window. A Python 3.8.8 implementation of the UMPA method available in reference [19] was used for T and DF recovery.

### 2.4. Image Analysis and Comparison

The contrast-to-noise ratio (CNR) is a widely used metric in X-ray imaging for distinguishing objects with different attenuation properties. Considering the advantages of T and DF imaging compared to BF imaging, it is logical to evaluate them using the same quantitative metrics. The CNR has been used in applications to X-ray microtomography using the single-shot speckle-based method for evaluation of T and phase contrast images in [20]. However, to the best of our knowledge, there are no publications that studied the CNR of T and DF images obtained with different reconstruction methods (XST, XSS and UMPA). The CNR was calculated according to the following definition:(5)$$CNR=\frac{\left|\bar{I}-\bar{I_{bg}}\right|}{\sigma_{bg}},$$
where $\bar{I}$ and $\bar{I_{bg}}$ refer to the intensities of the material and background (air) regions. To calculate both intensities, several regions of interest (ROIs) are defined along each material and in the region of air above object 4 (see Figure 2a). $\bar{I_{bg}}$ is calculated as the average of the means of the pixel values of each of the individual ROIs defined in the air region. $\sigma_{bg}$ is estimated as the square root of the quadratic sum of the standard deviations associated with each ROI divided by the number of ROIs. For each material, the CNR value and its uncertainty are calculated as the average and standard deviation of the CNR values obtained from the set of ROIs positioned in each material.

Since the number of images in the reconstruction differs between single-shot and scanning methods, we expect a corresponding difference in their CNR. Furthermore, variations in the T and DF signals across different materials also result in corresponding changes in CNRs.

The differences between CNR values of the different materials were statistically analyzed using the StatGraphics Centurion statistical package.

The linear absorption and diffusion coefficients, defined as in references [15,21],(6)$$\mu =\frac{\log T}{d},$$(7)$$\varepsilon =\frac{\log DF}{d},$$
where *d* is the thickness of the sample, are also used for quantitative analysis of the considered materials. Objects such as cork do not have completely uniform thickness; however, its variations are negligible compared with the inhomogeneity of the material.

## 3. Experimental Results and Discussion

First, we explore the information that is readily available in the BF images for the ODDs of 140 mm and 340 mm shown in Figure 2. A simple observation of these images allows us to easily distinguish materials 1 to 4 (see Figure 2b,c) due to their different attenuation properties, whereas objects 5 and 6 are barely visible. After flat-field and dark-current correction, differences in normalized BF images at both ODD are linked to variations in flat-field and dark-field images. The air STD for the BF image at 140 mm ODD is 0.7 times smaller than at 340 mm ODD.

Figure 3 displays the T and DF images recovered by the XST, XSS and UMPA using analysis window with *M* = 5 (Figure 3a,c) and *M* = 11 (Figure 3b,d). The images in Figure 3a,b correspond to ODD = 140 mm, while in Figure 3c,d, the images correspond to ODD = 340 mm. For simplicity, images and data obtained from them will be further referred to as T-XST, T-XSS and T-UMPA for the transmission modality and DF-XST, DF-XSS and DF-UMPA for the dark-field modality. As can be observed, T images, which are related to the materials’ absorption properties, follow the same trend as the BF images. In DF images, materials 4 and 5 (cork and expanded polystyrene) are well depicted with all the recovery methods, analysis windows and ODD. It can be seen that the single-shot method produces noisier T and DF images than the scanning methods. The images exhibit similar characteristics for both ODDs; however, they are noisier for ODD = 340 mm. The increase in the size of the analysis window reduces image noise, as observed when comparing Figure 3a–d, albeit at the cost of reduced resolution. The dependence of noise on these parameters will be clarified later.

The histograms in Figure 4 show that T images provide larger separation between pixel values than BF and DF images overall for the scanning methods due to the multiple images involved in the recovery (Figure 4c). In addition, the histograms clearly demonstrate the superior discrimination capabilities of the techniques using a large analysis window. This is particularly evident for T images across all methods, as well as for DF images in the scanning modalities. The increase in the window is also beneficial for XST. However, this increase has less impact on DF images except for the UMPA technique (Figure 4b).

For the quantitative characterization of the different objects in the sample, the average CNR values obtained from BF and T images are plotted versus the applied methods for both considered ODDs and analysis windows in Figure 5. It is observed that independently of the ODD and *M*, the magnitude of the CNR value follows the same trend for the objects in T and BF images. The largest CNR values correspond to nylon, followed by cardboard, cork, plastic, expanded polystyrene and foam for both. The results also show that the scanning methods (XSS and UMPA) clearly demonstrate superior material discrimination capabilities for T images for both considered ODDs and *M*. It is worth underlining that the CNRs in T-XSS and T-UMPA images are significantly larger than in T-XST and BF images. For all materials, the CNR values of the BF and T images for ODD = 340 mm are about twice smaller than the corresponding values obtained for ODD = 140 mm. The CNR significantly increases with *M*.

To explain the CNR behaviour we assume, as in reference [22], that the noise in T and DF images principally depends on the noise of raw data. We found that the CNR of T and DF images varies depending on the recovery method employed and the ODD. Specifically, the CNR of T images obtained via the single-shot method is comparable to that of BF images, whereas it is approximately four times higher for T images recovered using scanning methods. In this case, the noise is lower as more images (more photons) are used to recover the T images. Scanning methods utilize *N* = 15 images of the sample, which explains the approximately *N*^1/2^ times higher CNR compared to the XST and BF images. On the other hand, as ODD, and therefore magnification, increases, the X-ray fluence is distributed over a larger area and the CNR decreases approximately as SDD^−2^ (where SDD is the source–detector distance), which we approximately observe in Figure 5a–d. The CNRs of the T images obtained using a larger analysis window are almost twice as high and thus correlate with the ratio of the window sizes (11/5), which is approximately twice as high (see Figure 5a–d). 

CNR values for the DF images presented in Figure 6 follow a different order than for the T images regarding their magnitude with lower values (see ordinate axis) due to the higher image noise (about 10% higher $\sigma_{bg}$). DF images obtained by scanning methods present larger CNR values, as do T images. The largest CNR values in DF images correspond to cork (object 4) in all cases and the smallest ones to plastic (object 1) and foam (object 6). However, its variation with the ODD remains insignificant with a slight increase with ODD for low attenuating objects (5 and 6) and decrease for the other ones (like objects 1 and 2). We attribute this fact to the increase in the noise and to the decrease in the DF signal with the distance, as demonstrated in [18]. This decrease in the DF signal is material-dependent, providing different CNR behaviour with the ODD. While the scanning methods demonstrate the same CNR tendency, the values of the CNR for XSS and UMPA are slightly different, with better results for the UMPA method for *M* = 11.

Let us now analyze the uncertainties associated with the CNR estimations for the more reliable scanning method: UMPA. Figure 7 shows the plots of CNR values of DF images versus those of T images for the two different distances, (a,b) ODD = 140 mm and (c,d) ODD = 340 mm, and analysis windows, (a,c) *M* = 5 and (b,d) *M* = 11. For both distances, the CNRs of objects 3 and 4 in the DF images have very different values, in contrast to the corresponding values in the T images, which are very similar, as indicated by the error bars (standard deviation of the CNRs). On the other hand, the T images are crucial for discriminating materials with similar CNRs in the DF images, as is the case for objects 1, 2 and 3. In the case of the less attenuating objects (5 and 6), a difference in CNRs is observed for both distances, which is greater in the DF images than in the T images. Despite these inconsistencies, the discrimination capability of the combination of T and DF images is preserved for both ODDs, as follows from the analysis of the results in Figure 7.

Statistical analysis shows that there are significant differences between the CNR values of the different materials in the T images. In the DF images, the differences between the CNRs of objects 4 and 6 and the rest of the objects are statistically significant.

The linear absorption µ and diffusion ε coefficients are additional parameters that enable us to compare the effectiveness of the methods considered for material analysis. Figure 8a,c show that the values of μ are well differentiated for all objects, with the exception of objects 5 and 6. Additionally, the values for each object show a high degree of consistency across all considered methods, differing by less than 2%. The values of coefficients are independent on the analysis window and slightly dependent on the ODD. Comparing these values with the ones of linear attenuation coefficients calculated from BF images, we found that the latter were always higher for all the objects. This could be attributed to the DF signal that is also encoded into the BF signal. The μ values recovered from the measurements for both distances and analysis windows do not present significant differences.

On the other hand, the diffusion coefficients for the same material show significantly different values depending on the recovery method and the ODD. Upon detailed analysis of the uncertainties of the ε values recovered with each method, we conclude that the UMPA method provides ε values that are well differentiated (as observed in Figure 8b,d). The diffusion coefficient values are almost independent on the analysis window but significantly increase with the ODD.

Our findings partially corroborate the results of the simulations reported in reference [18]. These simulations demonstrated that the DF signal decreases with increasing ODD, and this reduction is dependent on the object’s material. Conversely, the T signal remains almost independent of the ODD. Our experimental results further validate that the variations in μ, and therefore the T signal, with ODD are minimal. Additionally, ε increases (DF signal decreases) as ODD increases (see Figure 8c,d). The variations in ε are indeed influenced by the object’s material.

In Figure 8e,f, the diffusion coefficient is presented against the attenuation coefficient for M = 11 and for both ODDs. The graphs show more clearly the behaviour of μ and ε with ODD observed in the other figures: a significant increase in ε with distance while the variations in μ remain within the range of uncertainties. Furthermore, the values of ε are significantly higher compared to those of μ, especially for larger ODD values. This facilitates improved material discrimination through the analysis of DF images. However, while the obtained ε values provide better differentiation, they cannot be used as universal material characteristics, unlike μ, which exhibits only small changes with ODD.

## 4. Conclusions

The analysis of T and DF images obtained through the three speckle-based methods, XST, XSS and UMPA, in a laboratory X-ray setup with a polychromatic microfocus source reveals that these methods offer additional information compared to BF images. The analysis of the histograms of the reconstructed T and DF images, as well as the CNR of the considered objects, demonstrates that the UMPA method is the most reliable recovery method compared to the other two studied speckle–based methods.

For implementation of the recovery methods, two analysis windows with side of *M* = 5 and *M* = 11 pixels were considered. It was demonstrated that the CNR of T and DF images is almost twice larger for the larger window due to proportional noise reduction. However, the values of the linear absorption (μ) and diffusion coefficients (ε) obtained from T and DF images recovered with UMPA slightly depend on the window size (Figure 8). This behaviour was found for all the applied methods. These results underline the robustness of the applied methods.

We found that the values of μ obtained from BF images were larger than ones recovered from T images (Figure 8a,c). On the other words, the BF signal is smaller than the T signal that can be attributed to scattering included in the BF signal.

It is worth mentioning that while the CNR values derived from T images decrease with increasing ODD due to noise, the CNR values obtained from DF images exhibit a more complex behaviour. These values may increase or decrease depending on the specific object. Additionally, we found that the linear diffusion coefficients increase with ODD, due to a reduction in the DF signal. This phenomenon can be explained by considering the recent study reported in [18], where numerical simulations of T and DF signals were conducted. The study demonstrated that the T signal remains unchanged, whereas the DF signal decreases with increasing ODD. Notably, the reduction in the DF signal was found to be material-dependent. Our findings align with these results. Specifically, the linear absorption coefficients are independent of ODD, while the linear diffusion coefficients increase with ODD in a material-dependent manner. Furthermore, the increase in noise associated with higher ODD can offset the decrease in the DF signal in terms of CNR for certain objects, which is consistent with our observations.

We conclude that linear absorption coefficients are more suitable for quantitative material characterization compared to linear diffusion coefficients, which are influenced by ODD. However, this does not diminish the usefulness of DF imaging. DF imaging enables the discrimination of materials with similar absorption properties through the comparison of CNR or ε values obtained at the same ODD. That said, determining an optimal ODD for this purpose is challenging due to the dependence of the DF signal on ODD, the materials being considered and the higher noise for larger ODD values.

We believe that our findings pave the way for wider applications of laboratory X-ray microfocus setups for material analysis.

## Figures and Tables

**Figure 1 sensors-25-02581-f001:**
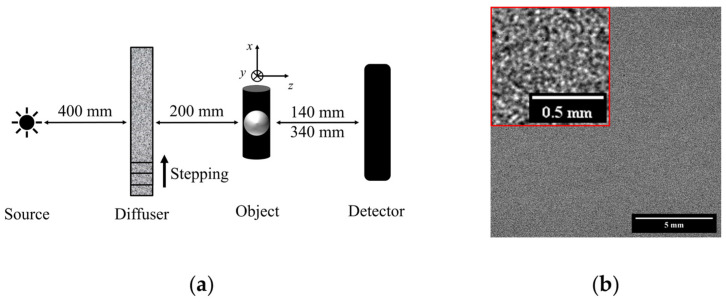
(**a**) Experimental setup. (**b**) Image of the reference speckle pattern formed by a sandpaper diffuser for an ODD of 340 mm. Inset shows a magnified view of the pattern.

**Figure 2 sensors-25-02581-f002:**
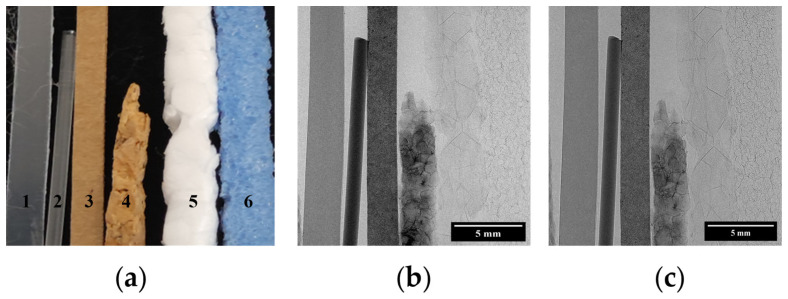
(**a**) Photograph of the studied sample containing six different materials: plastic (1), nylon (2), cardboard (3), cork (4), expanded polystyrene (5) and foam (6). (**b**) Bright field image of the sample for ODD = 140 mm. (**c**) Bright field image of the sample for ODD = 340 mm. The image for ODD = 140 mm has been cropped to obtain the same field of view for both distances.

**Figure 3 sensors-25-02581-f003:**
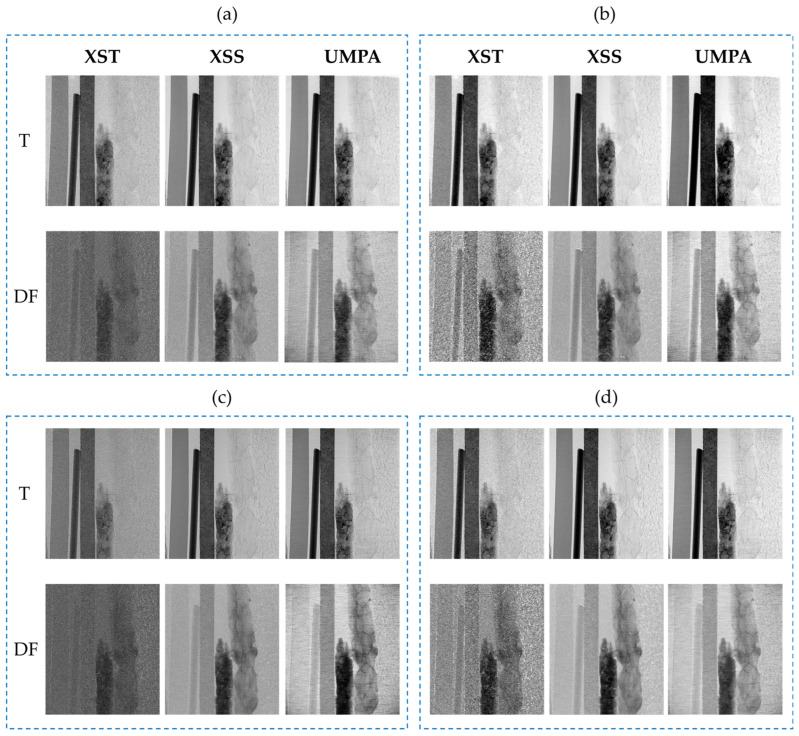
Transmission and dark-field images obtained for (**a**,**b**) ODD = 140 mm; (**c**,**d**) ODD = 340 mm. Results are shown for an analysis window of 5 × 5 (**a**,**c**) and 11 × 11 (**b**,**d**) pixels. The materials from left to right are plastic, nylon, cardboard, cork, expanded polystyrene and foam. The contrast of each image was selected for proper visual object distinction.

**Figure 4 sensors-25-02581-f004:**
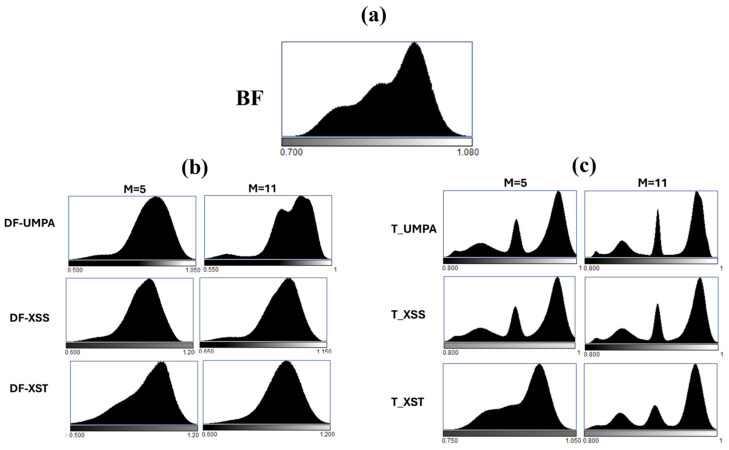
Histograms of the (**a**) bright field (BF) image obtained at ODD = 140 mm; (**b**) DF and (**c**) T images recovered with the three methods (UMPA, XSS and XST) for ODD = 140 mm and two analysis windows (*M* = 5 and *M* = 11).

**Figure 5 sensors-25-02581-f005:**
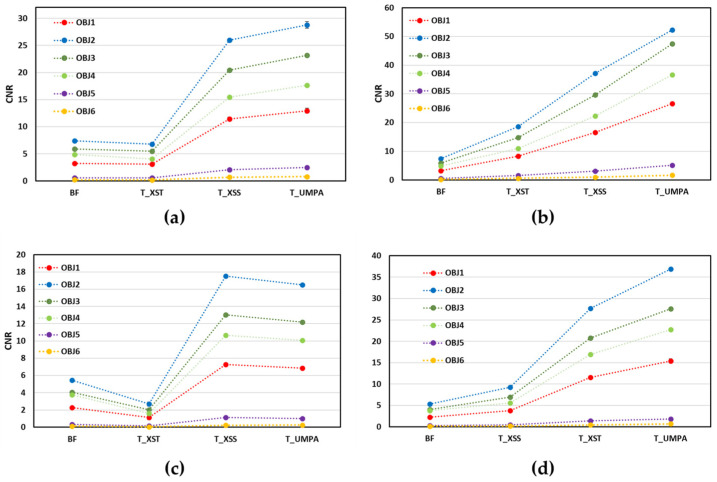
Average contrast-to-noise ratio of BF and T images retrieved with studied methods for all objects in sample. Graphics correspond to (**a**,**b**) ODD = 140 mm and (**c**,**d**) ODD = 340 mm. Analysis window equals *M* = 5 for (**a**,**c**) and *M* = 11 for (**b**,**d**). Symbols are red for plastic (OBJ1), blue for nylon (OBJ2), green for cardboard (OBJ3), green light for cork (OBJ4), purple for expanded polystyrene (OBJ5) and gold for foam (OBJ6). Dashed lines are included for eye guiding.

**Figure 6 sensors-25-02581-f006:**
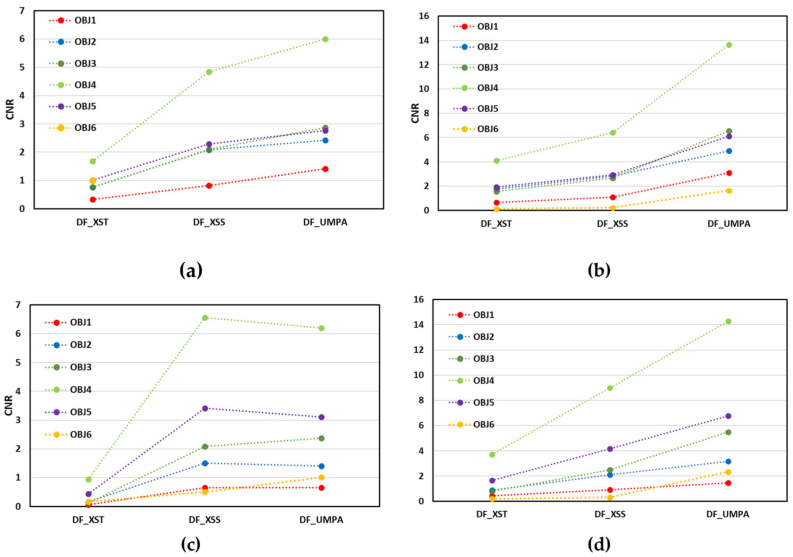
Average contrast-to-noise ratio of DF images retrieved with studied methods for all objects in the sample. Graphics correspond to (**a**,**b**) ODD = 140 mm and (**c**,**d**) ODD = 340 mm. Analysis window equals *M* = 5 for (**a**,**c**) and *M* = 11 for (**b**,**d**). Symbols are red for plastic (OBJ1), blue for nylon (OBJ2), green for cardboard (OBJ3), green light for cork (OBJ4), purple for expanded polystyrene (OBJ5) and gold for foam (OBJ6). Dashed lines are included for eye guiding.

**Figure 7 sensors-25-02581-f007:**
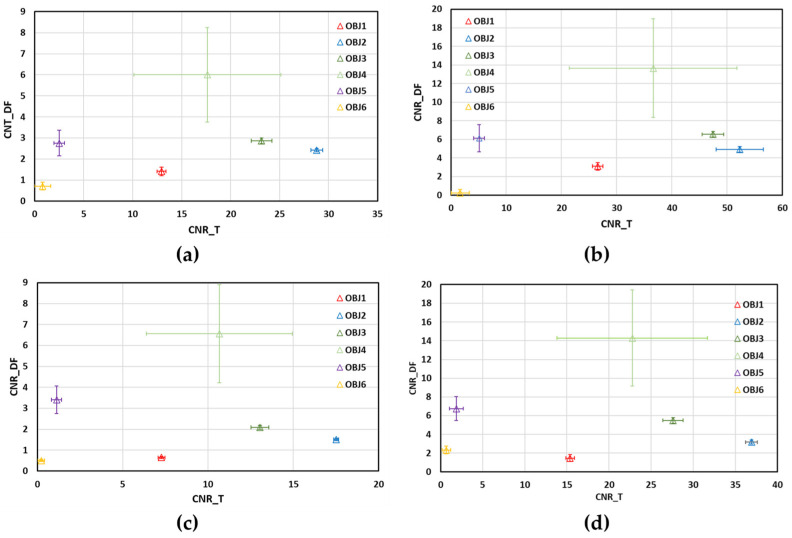
Contrast-to-noise ratio (CNR) values of DF images versus corresponding CNR values of T images recovered by UMPA for (**a**,**b**) ODD = 140 mm; (**c**,**d**) ODD = 340 mm and (**a**,**c**) *M* = 5; (**b**,**d**) *M* = 11. Symbols are red for plastic (OBJ1), blue for nylon (OBJ2), green for cardboard (OBJ3), green light for cork (OBJ4), purple for expanded polystyrene (OBJ5) and gold for foam (OBJ6).

**Figure 8 sensors-25-02581-f008:**
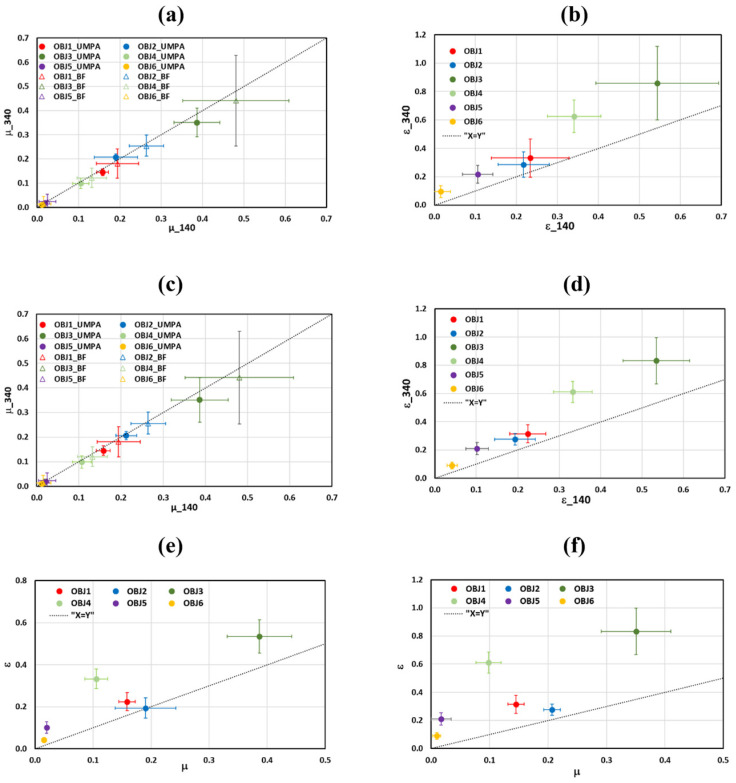
(**a**,**c**) Variation with ODD in the linear absorption coefficient (mm^−1^) recovered from T-UMPA images as well as the linear attenuation coefficient calculated from BF images. (**b**,**d**) Variations with ODD in the linear diffusion coefficient (mm^−1^) recovered from DF-UMPA images. The analysis window corresponds to *M* = 5 (**a**,**b**) and *M* = 11 (**c**,**d**). Diffusion coefficient versus attenuation coefficient for *M* = 11 and (**e**) ODD = 140 mm and (**f**) ODD = 340 mm. Symbols are red for plastic (OBJ1), blue for nylon (OBJ2), green for cardboard (OBJ3), green light for cork (OBJ4), purple for expanded polystyrene (OBJ5) and gold for foam (OBJ6).

**Table 1 sensors-25-02581-t001:** Thickness of materials considered.

Material	Plastic (1)	Nylon (2)	Cardboard (3)	Cork (4)	Expanded Polystyrene (5)	Foam (6)
Thickness (±0.05 mm)	0.6	0.9	0.4	1.4	1.9	2.4

## Data Availability

All the data supporting the findings of this study can be made available upon reasonable request.

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
