# Peer review of "Speckle-Based Transmission and Dark-Field Imaging for Material Analysis with a Laboratory X-Ray Source"

_sensors, 2025, doi:10.3390/s25082581_

Round 1
Reviewer 1 Report
Comments and Suggestions for Authors
The article "Speckle-based transmission and dark-field imaging for material analysis with a laboratory X-ray source" presents a approach to material analysis using X-ray speckle-based imaging techniques. It offers insights into the implementation and effectiveness of transmission (T) and dark-field (DF) imaging in a laboratory setting, comparing different image recovery methods and their impact on material discrimination. The article is well-written, logically structured. But To be honest, this paper has made no novel progress in methodology; it merely compares different methods. Moreover, it lacks in - depth data processing and material property investigation, which restricts the accuracy and universality of the methods as well as the scientific depth of the paper. It's hoped that the authors will incorporate more method comparisons in material property analysis or conduct more advanced data analysis to extract extra information.
I also have some suggestions as follow for revisions.
- In figure 2, the shape of sample 4 seems very different from the imaging results.
- Test usinga laboratory X-ray Please discuss the impact of coherence and intensity of light on the experimental results.
- Also, noticing that the scan step is even larger than the detector pixel, why not employ a smaller step to improve resolution?
- The use of contrast-to-noise ratio as a quantitative metric for material discrimination is justified and appropriate.However, the discussion of the results could be enhanced by including more direct comparisons with previous studies that have used similar metrics for evaluating X-ray imaging techniques.
- Have you experimented with other analysis window selections? Windows that are too small may introduce false information and lower the signal - to - noise ratio. There are relevant studies on this. Please discuss or compare the results of other window selections.
Reviewer 2 Report
Comments and Suggestions for Authors
This manuscript describes material analysis using a multi-contrast x-ray imaging setup, utilizing both x-ray attenuation and dark-field contrast using speckle technique. They compare the contrast to noise ratio in both x-ray contrast channels using three different phase-retrieval techniques. While there are a number of recent publications on this comparison, this paper focuses on using this signal for material analysis. I believe the topic is of interest to readers, however, unfortunately I do not believe that in its current state this manuscript is suitable for publication.
While the background and imaging setup are well described, I have issues with the methodology and results sections. The methodology incorporates contrast-to-noise ratio as a material discrimination metric, which is a confusing choice due to its dependence on imaging setup parameters. There are also a couple of results which do not match up with theory that lead me to believe that the methodology has been incorrectly implemented
I think there are a number of very interesting results that are not discussed. I would encourage the authors to resubmit after significant changes. In my view the paper should investigate the following questions:
- CNR for the three different methods, with either total exposure/pixel size/number of exposures kept consistent, which although not novel, is of interest with the compact setup used in this experiment, and would justify their later investigation
- Difference in absolute intensity for the three different methods (fig 6(a)(c), better labels/markers), and how this varies with distance. This is useful for obtaining “quantitative” measurements, that are potentially independent of setup – which the authors results show to not be possible
- Using one method (eg UMPA), looking at material discrimination using multi-model imaging (figure 5 with linear atten+diffusion rather than CNR on axes)
See below for my current concerns:
- Line71: The authors should define what they mean by “material discrimination”. To me this is differentiating between two materials based on their properties, which the authors seem to use CNR to do. In this case, references [15,16] do not do “material discrimination” and should be replaced by better references. [16] seems to only use a single material, and does not include transmission information. The following papers do not make use of speckle but do look into material discrimination, I leave it to the authors to choose appropriate references
- https://doi.org/10.1088/0031-9155/55/18/017
- https://doi.org/10.1103/PhysRevApplied.19.054042
- Line95: some justification for the smallest spot should be given. Spatial resolution is not important for this experiment. Is it due to the coherence requirements of speckle-based imaging? Do the authors expect a change in the results if a larger spot size was used?
- Section 2: overall very well described experimental setup
- Line 110: The visibility improves slightly with propagation distance, which I think needs to be explained. I would expect visibility to reduce due to source blurring
- Line 116: Bright field image should be better defined (seems to be a conventional radiograph). It is just unusual terminology to me
- Line 118: were the same flat fields and dark-current images used for BF and the three speckle methods? Important due to these potentially propagating noise into the images
- Line 122: Considering nylon is a type of plastic, please explicitly state what type of plastic is being used for material 1. Same with foam if this is known
- Section 2.3: It would be very useful to have the final image effective pixel size. The XST sounds like it has much larger pixels compared to the others (unless the window is a sliding window?), and XSS has non-square pixels?
- 3: following from this, the three methods, as well as BF, seem to have different statistics. I would like to see a total exposure time given for each approach.
- Section 2.3: Ideally, a metric should be given that accounts for both the total exposure time and pixel downsizing – something that gives the relative number of photons that have been used to form each image pixel. This is important to explain potential difference in noise between the techniques
- Line 155: is there an expected downside to using regular steps?
- Section 2.4: it is worth stating that the three techniques are known to be only semi-quantitative, and provide refernces, and we expect the “signal” (numerator on eq 5) to vary, not just the noise, particularly the dark-field
- Eq 4: I do not believe M has been defined, the number of pixels?
- Line 167: This is the one part of the methods that is very confusing. Was it one ROI in each material, and then multiple in air? Is sigma_bg from the quadratic sum of all ROI? Or only ROI in air (bg)? Or air and the ROI from the material of interest?
- Eq 6/7: only works if the material has uniform thickness within the ROI. Is this true for these samples?
- Line 185: the BF is flat-field corrected and hence should not have a lower intensity. This is very unexpected
- Line 196: why are they noisier? I assume because the lower statistics at this distance but this should be given.
- Figure 3: Labels would be very helpful here
- Figure3: is the “contrast” of the images selected for visualisation of each image? In other words, are the images displayed with the same colour axis?
- Figure 4: The legend (and throughout the manuscript) should be changed to the material names, “nylon” rather than “obj 2”. It is very hard to follow as it is.
- Line 226: some discussion on why objects 2/3/5 separate in CNR when the ODD distance increases is required. It is not obvious to me why this is the case
- Line 228: again, why does the CNR change significantly with ODD for XST? But not the others
- Line 229: “sensible” should probably be “sensitive”
- Figure 5: To me, CNR is a really strange metric used to differentiate materials. Why not just intensity? The noise component just means the measurement is affected by the exposure time, retrieval method, system geometry. Yes CNR is useful to judge whether you can make it out from the background, but this can be included with the error bars, such as figure 6
- Figure 5: From here, including XSS just adds confusion, I advise just sticking to UMPA which seems to have better CNR in nearly every case
- Figure5 : I do not believe the insets are necessary, and they are not very readable either
- Line 260+298: again, why is mu different for the different ODD in BF? This seems incorrect
- Figure 6: (a) doesn’t seem to have the three different methods plotted. (c) does have the different methods, but we cannot tell which is which. (d)(b) both seem to be repeated plots but with error bars. This figure needs redone
- Line 291: Why is CNR higher for speckle compared to BF? Is it down to longer exposures? Would removing the sandpaper and exposing for longer give better CNR than speckle?
- Line 304: I think this is an important finding and one which should be emphasised in the abstract. I do not even see it in the results
Round 2
Reviewer 2 Report
Comments and Suggestions for Authors
I commend the authors for their thorough response which has improved the manuscript significantly and addressed all concerns. I look forward to their further work